# The influence of stimulus duration on olfactory perception

**Praveen Kuruppath**[¤a]*, **Leonardo Belluscio**[¤b]

Developmental Neural Plasticity Section, National Institute of Neurological Disorders and Stroke, National Institutes of Health, Bethesda, Maryland, United States of America

¤a  Current address: Department of Physiology and Biophysics, University of Colorado School of Medicine, Aurora, Colorado, United States of America
¤b  Current address: Howard Hughes Medical Institute, Chevy Chase, Maryland, United States of America
* praveen.kuruppath@cuanschutz.edu

## Abstract

The duration of a stimulus plays an important role in the coding of sensory information. The role of stimulus duration is extensively studied in the tactile, visual, and auditory system. In the olfactory system, temporal properties of the stimulus are key for obtaining information when an odor is released in the environment. However, how the stimulus duration influences the odor perception is not well understood. To test this, we activated the olfactory bulbs with blue light in mice expressing channelrhodopsin in the olfactory sensory neurons (OSNs) and assessed the relevance of stimulus duration on olfactory perception using foot shock associated active avoidance behavioral task on a "two-arms maze". Our behavior data demonstrate that the stimulus duration plays an important role in olfactory perception and the associated behavioral responses.

**Data Availability Statement:** All relevant data are within the paper and its Supporting information files.

**Funding:** This work was supported by the Intramural Research Program of the National Institutes of Health–National Institute of

## Introduction

Stimulus duration is an important parameter of the sensory stimuli. The significance of stimulus duration has been extensively studied in the visual, auditory and tactile sensory system [1–4]. In the olfactory system, the role of stimulus duration is not well understood because of the difficulty controlling the input duration which is affected by sniffing and the complex interactions between odorants and the epithelium. Rodents in their natural environments encounter mixtures of odorants of diverse quantities, qualities and complexities. The interaction of these complex odor mixtures and the activation of different OSNs shows antagonistic interactions among odorants [5, 6].

Sniffing is precisely and strongly modulated by task demands, behavioral state, and stimulus context. In vertebrates, the olfactory epithelium is enclosed deep in the nasal cavity, which requires the inhalation of air for odorants to reach the olfactory sensory neurons (OSNs). Activation of the OSNs can only occur during resting respiration or by the voluntary inhalation of air in the context of odor-guided behaviors, such as sniffing. Thus, for a particular odor task or a context, sniffing patterns shows a particular strategy for olfactory sampling [7–9].

Neurological Disorders and Stroke (1ZIANS003116-01). The funders had no role in study design, data collection, and analysis, decision to publish, or preparation of the manuscript.

**Competing interests:** The authors have declared that no competing interests exist.

**Abbreviations:** OB, Olfactory bulb; OMP, Olfactory marker protein; OSNs, Olfactory sensory neurons.

Previous studies have demonstrated the behavioral relevance of precise olfactory timing relative to sniffing [10–13]. Mice discriminated simple odors and binary mixtures equally well with an accuracy of 95%. However, mice took more time to discriminate the binary mixtures than simple odors suggesting that odor discrimination is timing and stimulus dependent [14]. Physical and temporal properties of the stimulus are also important for obtaining information about the odor plumes when the odor is delivered into the environment [15–20]. For mammals and insects, temporal information serves as an important cue for finding the odor source in their natural environment to locate food, mates, and avoid predators [21–30].

For olfaction, temporal information can be significant for identifying an odor. Previous studies in honeybees by Wright and co-workers observed that increasing the sampling time improves their ability to recognize and differentiate odors [31]. Their results show that sampling time affects olfactory learning, recognition, and discrimination, suggesting that having a longer time to sample an odor stimulus yields more information about the odor's presence and the molecular identity, which improves the ability of the olfactory system to form a neural representation of an odor's molecular identity. Another study by Szyszka et al., shows that honeybees can detect temporal incoherence between odorant stimuli in millisecond range and use this information to extract odorant identity [32]. However, no studies have explored the behavioral relevance of stimulus duration in the odor identity in mammals, due to the fact that input duration is widely affected by sniffing. To reduce the variability in odor stimulation, in many studies odor is delivered at a certain phase of the respiration cycle [33–37].

In this study, we controlled the sensory inputs to the unilateral and bilateral olfactory bulbs (OB) with blue light in tetO-ChIEF-Citrine mouse line, where channelrhodopsin-2 is expressed in all the olfactory sensory neurons (OSNs) and studied the behavioral responses to the light pulse stimulation with different stimulus durations in unilateral and bilateral OB. We found that mice respond differently to shorter and longer stimulus durations, suggesting that olfactory information changes with the stimulus duration.

## Materials and methods

### Experimental animals

All animal procedures conformed to National Institutes of Health guidelines and were approved by the National Institute of Neurological Disorders and Stroke Institutional Animal Care and Use Committee. Mice were bred in-house and were maintained on a 12 h light/dark cycle with food and water ad libitum.

The tetO-ChIEF-Citrine line was generated from pCAGGS-I-oChIEF- mCitrine-I-WPRE (7.7kb; Roger Tsien, UCSD), which contains the coding sequence for mammalian-optimized ChIEF fused to the yellow fluorescent protein Citrine, at the National Institute of Mental Health Transgenic Core Facility (Bethesda, MD) as previously described [38, 39]. The OMP-tTA knock-in mouse line expressing the tetracycline transactivator protein (TTA) under the control of the OMP-promoter was a gift from Dr. Joseph Gagos. Experimental animals were OMP-tTA+/- / tetO-ChIEF-Citrine+/- (OMP-ChIEF), generated by crossing heterozygous tetO-ChIEF-Citrine (tetO-ChIEF-Citrine+/-) with homozygous OMP-tTA mice (OMP-tTA-/-).

### Genotyping

OMP-ChIEF pups were identified by the visualization of fluorescence in the nose and OB of P0-P2 pups under epifluorescence illumination.

## Animal preparations

Data was collected from 12 OMP-ChIEF mice. Experimental animals were prepared as described previously [40, 41]. Briefly, mice were anesthetized with an intraperitoneal injection of Ketamine/Xylazine mixture 100 and 10 mg/kg body weight, respectively. Each animal was fixed with a stereotactic frame with the head held in place by a bar tie to each temporal side of the skull. The animals were kept warm with hand warmers (Grabber, Grand Rapids, MI, USA). Surgery was started when the animal showed no movement in response to foot pinching. A craniotomy was performed above the skull over each OB. Fiber optic pins were implanted on the dorsal surface of each OB as described previously [40, 41]. Mice were injected with Ketoprophen (5mg/kg) immediately after the surgery. Animals were allowed to recover in their home cage for one week.

## Behavioral procedure and training

**Light stimulation and foot shock avoidance training.** Behavioral training began one week after the animals recovered from the surgery. Training was performed for two days on a "two-arms maze", modified from 'Y' maze with two equally sized open arms and one permanently closed arm. Each open arm was independently paved with an electric grid shock floor. The mice were connected to a 400 μm core-diameter optical fiber attached to a 473 nm solid-state variable-power laser (LaserGlow Technologies, Toronto, Canada, Fig 1A) and allowed to habituate in the "two-arms maze" for 15 minutes. The time spent in each arm of the "two-arms maze" was recorded and assessed for arm preference. The preferred arm for the mice was selected as the Light zone and the opposite arm served as the Safe zone. If the mice did not demonstrate a preference for either arm, the Light zone was randomly selected. After the habituation, mice were re-introduced to the "two-arms maze" for the ten-minute training

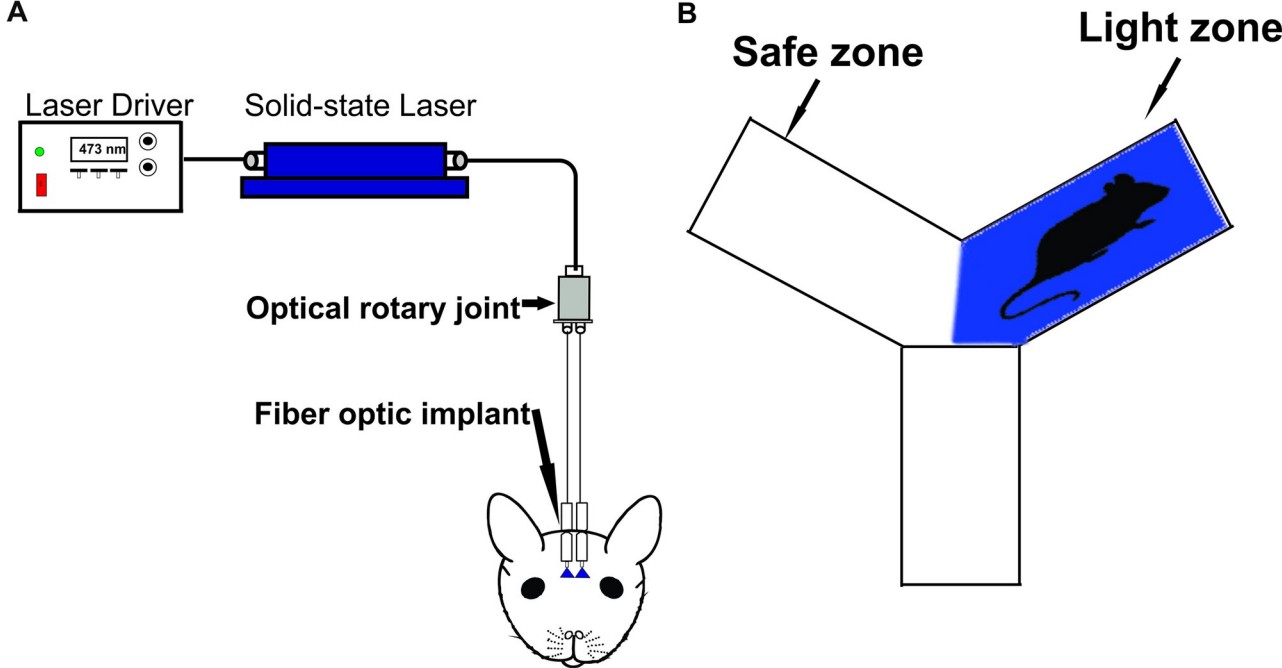

**Fig 1. Optogenetic stimulation of olfactory bulb.** A, Schematic diagram of the functional system. B, Behavioral setup. Light zone indicates the area where the light stimulation is delivered. Safe zone indicates the area where mice can escape from the light stimulation and foot shock.

session. A ten-minute reinforcement training was performed every day 60 minutes prior to the test session to enhance learning. The training included light stimulation followed by a mild foot shock. The foot shock avoidance training is paired with either left or right OB stimulation. The light stimulation and the foot shock were delivered in the Light zone each time when the mice completely entered that zone (Fig 1B). The mice had free access to the Safe zone to escape the foot shock. Light stimulation, consisting of a train of ten light pulses of 50 ms duration with an interval of 150 ms, was externally triggered by a Master-8 timer (A.M.P.I, Jerusalem, Israel) [42–44]. The output power of the light pulses was measured and adjusted to 20–22 mw with reference to previous studies [45–48]. The mild foot shock (0.65mAmps, 5 s) generated by a stand-alone shock generator (Med Associates, USA) was delivered 2 s after the light stimulation by Master-8 timer. The mice were trained to move to the Safe zone when the light stimuli and the foot shock were delivered in the Light zone.

**Light zone avoidance test.** The Light zone avoidance test was performed for two days. A reinforcement training was performed each day before the testing. Test stimuli consisted of a train of ten light pulses of 10 or 25 ms duration with an interval of 150 ms. The testing began 60 minutes after the reinforcement training. The light stimulation was delivered in the Light zone each time when the mice completely entered that zone. Before the testing, the electric grid shock floor was removed from the "two-arms maze", so the mice did not experience foot shock during the test session. The activity of the mice in the "two-arms maze" was evaluated in blocks of three trials and each trial was 15 minutes. In the first trial, the mice could explore the arena without any light stimulation and the baseline behavior was assessed. The time spent in each arm was calculated to evaluate arm preferences after the foot shock training session. For the following trials, the Light zone and Safe zone were selected, as indicated previously. The time spent in each arm was calculated by tracking the animal's movement in the "two-arms maze" using Any-maze video tracking software (Stoelting, IL, USA). A heatmap was also generated for each trial which displays the amount of time mice spent in different parts of the arena. A range of colors indicates the total time spent in the area with blue indicating the shortest and red as the longest time.

## Statistical analysis

All statistical analyses were done by Graph Pad prism software (Graph-pad, San Diego, USA). Statistics are displayed as mean ± SEM. Paired t test was used for the comparison. Differences were determined significant when $p < 0.05$.

## Results

### Olfactory information changes with stimulus duration

Duration is an important input parameter of olfactory stimuli [43] and the temporal properties of the stimulus are key for obtaining necessary information about the identity, intensity, and the direction of the olfactory stimuli from the environment [21–23, 25]. While previous studies suggest a substantial role of early-responding primary glomeruli and response latencies in odor identification [49, 50], a direct connection between stimulus duration and odor identity is still lacking. To test whether stimulus duration influence the olfactory information, we optically stimulated ChIEF expressing OSNs in olfactory bulb and assessed the behavior in response to a pattern of 10 light stimuli with a duration of 10 or 25 ms with an inter-stimulus interval (ISI) of 150 ms. After training, we tested the Light zone avoidance response to OB stimulation, linked to the foot shock. First, we tested the mouse baseline behavior and calculated the time spent in each arm by allowing the mice to freely explore the arena. The aim of the baseline behavior analysis was to confirm if the mice have a preference to a particular arm

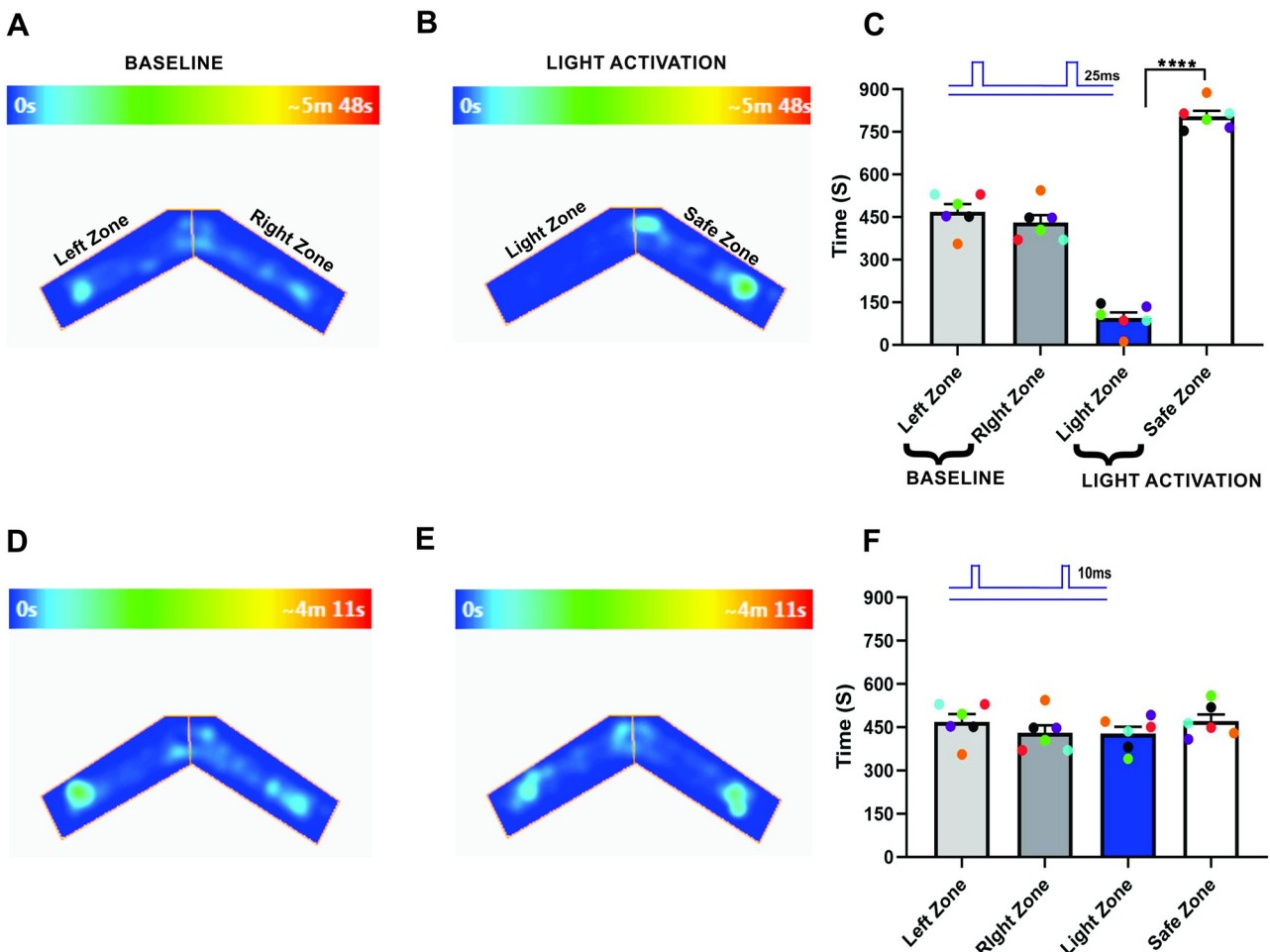

**Fig 2. Stimulus duration changes the olfactory information.** A, B, an example of heat-map showing the animal's position in the "two-arms maze" during baseline (A) and 25 ms unilateral light stimulation (B). C, Average amount of time explored in each zone in baseline (Left zone, Right Zone) and 25 ms unilateral light stimulation trials (Light zone, Safe zone). D, E, Heat-map of mouse position during baseline (D) and 10 ms unilateral light stimulation (E). F, Average amount of time spent in each zone in baseline and 25 ms unilateral light stimulation trials. (****P<0.0001, n = 6 animals, Heat map generated by ANY-maze version 5.2, https://www.anymaze.co.uk/index.htm).

of the "two-arms maze" after the foot shock training. Our baseline data show that the mice spent nearly equal amounts of time in both arms of the "two-arms maze" (Left zone— 469.3 ± 26.73 s, Right zone- 430.7 ± 26.73 s, P = 0.50, Fig 2A, S1 Movie, n = 6). Then, we unilaterally stimulated the OB with 25 ms duration light pulses with an interval of 150 ms. We found that the mice avoided the Light zone during 25 ms OB stimulation and spent most of their time in the Safe zone (Light zone—94.75 ± 19.47 s, Safe zone—805.3 ± 19.47 s, P = <0.0001, Fig 2B and 2C, S2 Movie, n = 6). We also tested whether the avoidance response to the light stimulation was equally probable in both arms of the "two-arms maze". To confirm this, in some trials we delivered light stimulation when the mice reached the Safe zone. We found that the mice avoided the Safe zone during the light stimulation, indicating that the avoidance response is clearly linked to olfactory information, but not to spatial information.

Next, we stimulated the OB with 10 ms duration light pulses. Here, we found that the time spent in each arm during the baseline behavior trial (Left zone—469.3 ± 26.73 s, Right zone—430.7 ± 26.73 s, P = 0.50, Fig 2D, n = 6) and the light activation trial (Light zone—

428.5 ± 23.26 s, Safe zone—471.5 ± 23.26 s, P = 0.40, Fig 2E) were similar, indicating that the mice did not identify the foot-shock linked olfactory information. Thus, they did not avoid the Light zone during the 10 ms light pulses (Fig 2F, S3 Movie, n = 6).

In agreement with previous study in Drosophila [51], our results suggest that the duration of the stimulus changes the olfactory information.

## Bilateral input duration alters the unilateral olfactory information

How an animal identifies a specific odor is one of the significant challenges for the olfactory system. Olfactory perception and behavior depend mainly on the ability to identify an odor across a wide variety of odor mixtures [12, 14, 52, 53]. Previous studies report that bilateral olfactory input enhances navigation and chemotactic behavior [25, 54]. To assess if the duration of the bilateral stimulus influenced olfactory perception, we stimulated each OB with different duration stimuli on unilaterally trained mice and assessed if the mice could identify the foot-shock linked olfactory stimulus from the synchronized bilateral OB stimulus. We found that during baseline behavior trials, mice visited both arms equally (Left zone—434.7 ± 58.81 s, Right zone—465.3 ± 58.81 s, P = 0.80, Fig 3A), and when we synchronously stimulated each

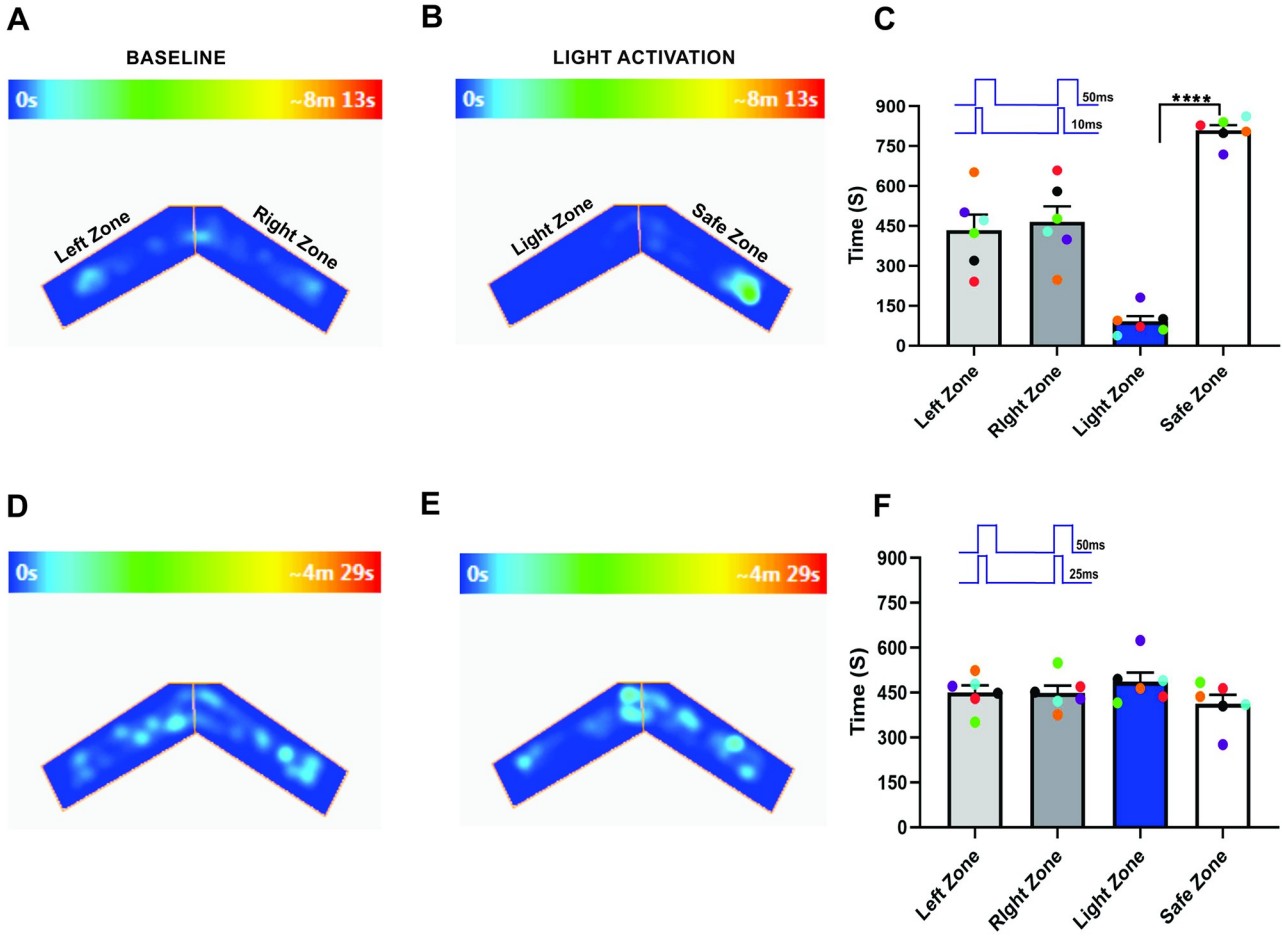

**Fig 3. Bilateral input duration changes the unilateral olfactory information.** A, B, an example of heat-map showing the animal's position in the "two-arms maze" during baseline (A) and dual OB (50–10 ms) light stimulation (B). C, Average amount of time explored in each zone in baseline and dual OB (50–10 ms) light stimulation trials. D, E, Heat-map of mouse position during baseline (D) and dual OB (50–25 ms) light stimulation (E). F, Average amount of time spent in each zone in baseline and dual OB (50–25 ms) light stimulation trials. (****P<0.0001, n = 6 animals).

OB with 50 and 10 ms light pulses, mice avoided the Light zone (Light zone—91.08 ± 20.30 s, Safe zone—808.9 ± 20.30 s, P = <0.0001, Fig 3B and 3C, n = 6). This result indicates that glomeruli activated during short duration light stimulation on the contralateral OB are insufficient for influencing the ipsilateral olfactory information. Next, we synchronously stimulated each OB with 50 and 25 ms light pulses. Here, we found that the time spent in each arm during baseline behavior trials (Left zone—450.3 ± 23.74 s, Right zone—449.7 ± 23.74 s, P = 0.99, Fig 3D) and light activation trials (Light zone—487.4 ± 29.99 s, Safe zone—412.6 ± 29.99 s, P = 0.27, Fig 3E and 3F, n = 6) was almost equal and they did not avoid the Light zone. This suggests that the mice did not identify the foot-shock linked olfactory information. Our results suggest that mice integrate the olfactory information from the contralateral OB during synchronized bilateral stimulation at longer stimulus duration and perceive it as a different olfactory information. During the shorter stimulation, identity of the olfactory information remains same. Together, these results suggest that the duration of the bilateral input influences olfactory information.

To confirm this, we performed light stimulation and foot-shock avoidance training on a new group of animals (n = 6). Here, we trained the mice with 50-ms light pulses applied simultaneously to both olfactory bulbs and paired with foot shock. After the training, we stimulated each OB with different duration light pulses and tested the response of mice in the Light zone avoidance test. Our results show that when each olfactory bulb was synchronously stimulated with 50 and 25 ms light pulses, mice avoided the Light zone. This result indicated that mice can identify the bilaterally-applied olfactory stimulus linked to the foot-shock (Left zone—455.8 ± 33.16 s, Right zone—444.2 ± 33.16 s, P = 0.87, Light zone—88.08 ± 19.28 s, Safe zone—811.9 ± 19.28 s, P = <0.0001, Fig 4A–4C, S4 Movie, n = 6). We then tested to determine if the mice can discriminate the foot-shock linked olfactory information when exposed to the stimulus duration of 50 and 10 ms. We found that mice did not avoid the Light zone and continued to stay in the Light zone (Left zone—468.5 ± 35.70 s, Right zone—431.5 ± 35.70 s, P = 0.63, Light zone—468.3 ± 26.24 s, Safe zone—431.8 ± 26.24 s, P = 0.52, Fig 4D–4F, S5 Movie, n = 6), suggesting that mice did not identify the bilaterally-applied stimulus when one bulb is stimulated with shorter duration light pulses. This result confirms our previous results, that the bilateral input duration influences the olfactory information.

Next, we tested whether mice perceive a change in the stimulus when both OBs are stimulated for the same duration. Here, we synchronously stimulated both OBs with 25-ms light pulses. Our results show that mice avoided the light zone during 25-ms light stimulation (Left zone—430.7 ± 28.60 s, Right zone—469.3 ± 28.60 s, P = 0.53, Light zone—152.8 ± 34.03 s, Safe zone—747.3 ± 34.03 s, P = 0.0003, Fig 5A–5C, S6 Movie, n = 6). We also simultaneously stimulated each OB with 10-ms light pulses and tested whether the mice detect the foot shock-linked olfactory stimulus during the short dual bulb stimulation. We found that, similar to the baseline condition, the mice spent almost equal amounts of time in both arms of the "two-arms maze" (Left zone—474 ± 46.85 s, Right zone– 426 ± 46.85 s, P = 0.63, Light zone—475.8 ± 45.38 s, Safe zone—424.3 ± 45.38 s, P = 0.59, Fig 5D–5F, S7 Movie, n = 6). Together, these results confirm that stimulus duration influences olfactory information. To verify that the ChIEF expressed in the OSNs is getting activated during the 10-ms light pulse stimulation, we trained the mice with light stimulation to both OB simultaneously with 10 ms duration on each OB and paired it with foot shock. After training, we tested the avoidance behavior of the mice. We found that the mice avoided the Light zone, indicating effective activation of OSNs during the shorter duration light pulses (Left zone—412.8 ± 30.32 s, Right zone—487.3 ± 30.32 s, P = 0.30, Light zone—204 ± 15.30 s, Safe zone—696 ± 15.30 s, P = 0.0005, Fig 5G–5I, n = 4). A previous study by Li et al. (2014) also showed that mice can discriminate the activation of ChIEF with 10 ms light pulses [43].

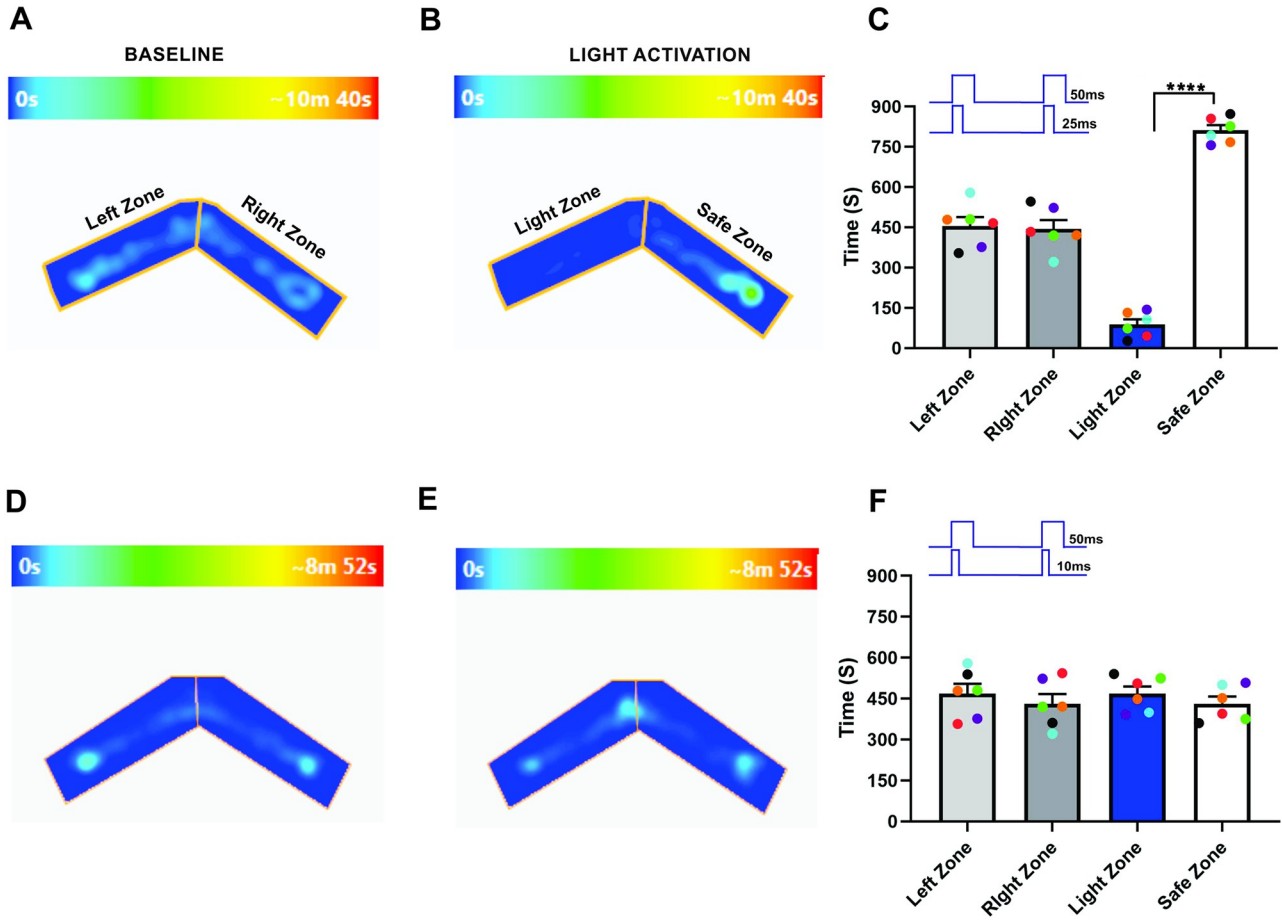

**Fig 4. Stimulus duration influences the bilateral olfactory information.** A, B, an example of heat-map showing the animal's position in the "two-arms maze" during baseline (A) and dual OB (50–25 ms) light stimulation (B). C, Average amount of time explored in each zone in baseline and dual OB (50–25 ms) light stimulation trials. D, E, Heat-map of mouse position during baseline (D) and dual OB (50–10 ms) light stimulation (E). F, Average amount of time spent in each zone in baseline and dual OB (50–10 ms) light stimulation trials. (****P<0.0001, n = 6 animals).

Finally, to verify that the observed responses from light stimulation were the result of activation of ChIEF-expressing neurons and not from the use of light as a visual cue, we used green light (540nm, output power, 20-22mw), which does not activate ChIEF; mice are also relatively insensitive to such long wavelength light [55–57]. During the green light stimulation, we did not observe significant behavioral differences from the baseline behavior (Left zone—457.5 ± 51.09 s, Right zone—442.5 ± 51.09 s, P = 0.89, Light zone—463.8 ± 16.87 s, Safe zone—436.2 ± 16.87 s, P = 0.45, Fig 6A–6C, n = 6), confirming that the mice did not use visual cues to perform the task.

Together, our results demonstrate that the duration of an olfactory stimulus plays an important role in its discrimination. Animals might detect shorter-duration olfactory stimuli but require longer stimuli in order to identify and generalize olfactory information.

## Discussion

Temporal properties of the stimuli are key to obtaining information from odor plumes in the environment [21, 22]. Previous studies have provided evidence that animals can use the temporal properties of an olfactory stimulus to gather information regarding the spatial

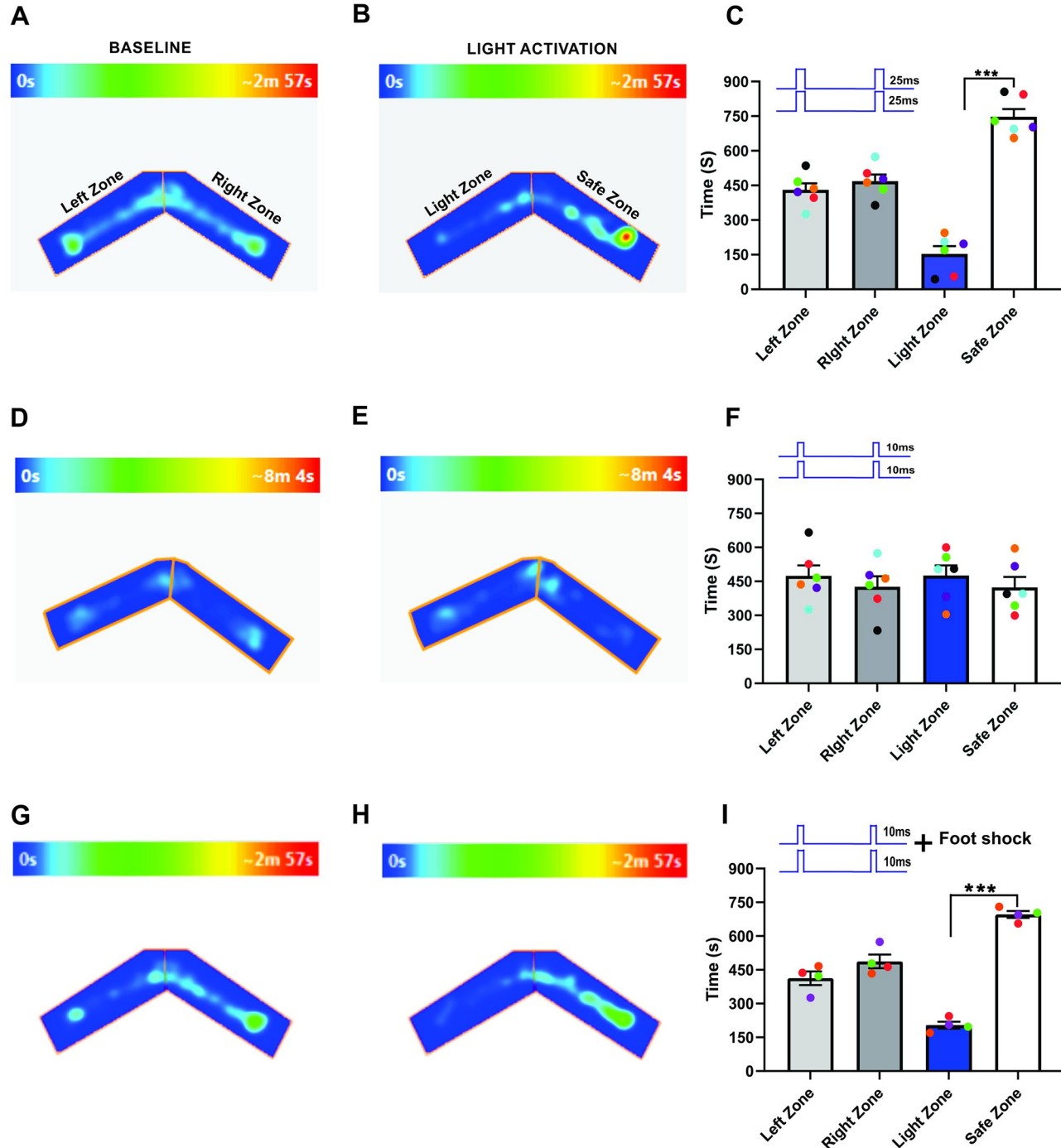

**Fig 5. Bilateral olfactory information.** A, B, an example of heat-map showing the animal's position in the "two-arms maze" during baseline (A) and dual OB (25–25 ms) light stimulation (B). C, Average amount of time explored in each zone in baseline and dual OB (25–25 ms) light stimulation trials. D, E, Heat-map of mouse position during baseline (D) and dual OB (10–10 ms) light stimulation (E). F, Average amount of time spent in each zone in baseline and dual OB (10–10 ms) light stimulation trials. (***P0.0003, n = 6 animals). Glomerular activation during shorter duration stimulus. G, H, an example of heat-map showing the animal's position in the two-arms maze during baseline (G) and 10 ms bilateral light stimulation (H). I, Average amount of time explored in each zone in baseline and 10 ms bilateral light stimulation trials (***P0.0005, n = 4 animals).

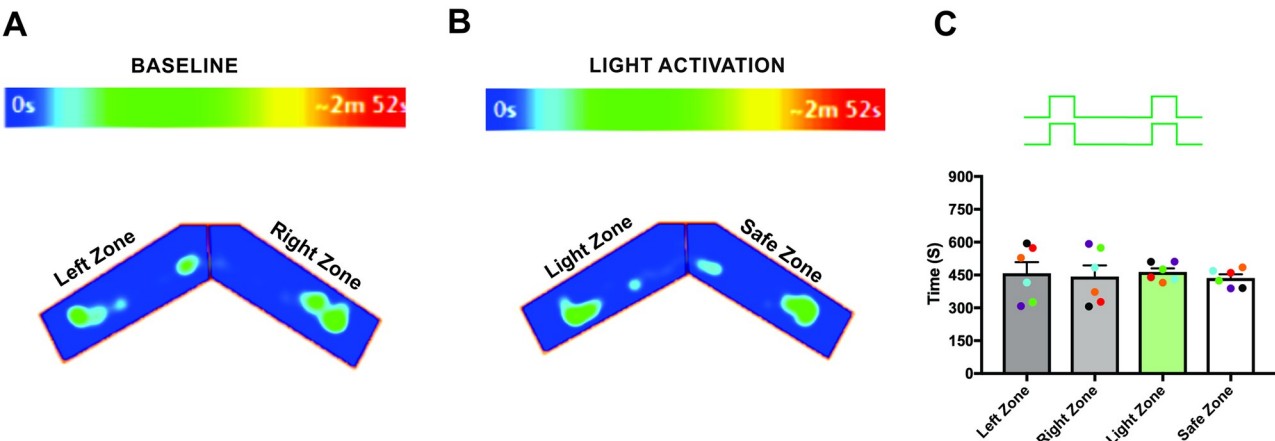

**Fig 6. Green light stimulation did not activate the olfactory system of OMP-ChIEF mice.** A, B, Heat-map of mouse position during baseline (A) and green light stimulation (B). C, Average amount of time spent in each zone during green light stimulation (n = 6).

localization of an odor source [9, 26–30, 43]. The role of stimulus timing in olfaction however remains poorly understood because of the difficulty in precisely controlling an odorant stimulus and the complex sniffing pattern of the animals. Taking advantage of the transgenic mice, our study revealed that the duration of the olfactory stimuli carries information about the identity of an olfactory stimulus. A previous study in Drosophila has shown that flies can generalize an odor from a certain range of concentration, but when the concentration falls below certain threshold, the same odor will be detected as different [51]. It is well-established that the number of activated glomeruli increases alongside the odor concentration, making the odor generalization possible [50].

Sniffing is a characteristic odor sampling behavior in rodents, accompanied with an increase in both nasal airflow rate and breathing frequency. In rodents, identity and the generalization of an olfactory information can be influenced by the air flow rate, sniffing, and complex interaction of the odorant molecule with the olfactory epithelium. A previous study found that the glomerular responses are highly influenced by the nasal airflow rate, but not by sniffing frequency, suggesting that nasal airflow rate, but not respiratory frequency, is a key factor that regulates olfactory sensitivity of the glomeruli [58].

Precise control of such parameters makes it difficult to study the role of stimulus duration in odor identity. Taking the advantage of optogenetics, Li et al., behaviorally showed that mice expressing ChR2 in olfactory sensory neurons can discriminate stimulus duration with a resolution of 10 ms [43]. There are no other reports available that demonstrate the role of stimulus duration on olfactory identity. Previous studies in invertebrates have shown that they can encounter brief pulses of odor, and these pulses may contain valuable information about odor location, concentration, and odor identity [21, 22, 31].

In our study, we took the advantage of ChIEF variant of ChR2 and optogenetics to precisely control the activation of glomeruli to study the role of stimulus duration on olfactory identity. Selection of correct ChR2 variants with the right properties largely influences the precision and efficiency of light-induced depolarization in the neurons. The recovery from the desensitization from both repetitive and prolonged light stimulation is also important in characterizing ChR2 variants correctly. The main characteristic feature of the ChIEF is Improved membrane trafficking and minimal desensitization with long and pulsed stimulation and the limitations includes lower light sensitivity and incomplete recovery from the desensitized response [39,

45–47]. With the precise and controlled glomerular activation, our study provides evidence for the significance of stimulus duration on olfactory identity in vertebrates.

Further research is required to fully understand how the stimulus duration influences the olfactory cortical neurons in encoding olfactory information for odor identity and localization.

In natural environments, animals confront more complex problems such as having to identify the quality and complexity of an odor mixture. The use of suitable physiological and psychophysical paradigms will be a crucial step for further understanding the complexity of neural coding in the olfactory system.

## Conclusion

Taking advantage of optogenetics and behavior test, the present study demonstrates that stimulus duration plays an important role in identifying and generalizing olfactory information. Our study shows that mice respond differently to shorter and longer stimulus durations, suggesting that the olfactory information changes with the stimulus duration.

## Supporting information

**S1 Movie. Mouse baseline exploratory behavior.**
(MP4)

**S2 Movie. Unilateral foot shock trained mice avoiding Light zone during 25 ms olfactory bulb stimulation.**
(MP4)

**S3 Movie. Unilateral foot shock trained mice did not avoid Light zone during 10 ms olfactory bulb stimulation.**
(MP4)

**S4 Movie. Bilateral foot shock trained mice avoiding Light zone during longer synchronized bilateral olfactory bulb stimulation (50 & 25 ms).**
(MP4)

**S5 Movie. Bilateral foot shock trained mice did not avoid Light zone during shorter synchronous bilateral olfactory bulb stimulation (50 & 10 ms).**
(MP4)

**S6 Movie. Bilateral foot shock trained mice avoiding Light zone during synchronized bilateral olfactory bulb stimulation (25 & 25 ms).**
(MP4)

**S7 Movie. Bilateral foot shock trained mice did not avoid Light zone during synchronized bilateral olfactory bulb stimulation (10 & 10 ms).**
(MP4)

## Author Contributions

**Conceptualization:** Praveen Kuruppath, Leonardo Belluscio.

**Data curation:** Praveen Kuruppath.

**Formal analysis:** Praveen Kuruppath.

**Funding acquisition:** Leonardo Belluscio.

**Investigation:** Praveen Kuruppath.

**Methodology:** Praveen Kuruppath.

**Project administration:** Praveen Kuruppath.

**Supervision:** Praveen Kuruppath.

**Validation:** Praveen Kuruppath.

**Visualization:** Praveen Kuruppath.

**Writing – original draft:** Praveen Kuruppath.

**Writing – review & editing:** Praveen Kuruppath.

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
