## [Decision Letter · Decision Letter 0]

23 Oct 2020

PONE-D-20-30938

The influence of stimulus duration on olfactory identity

PLOS ONE

Dear Dr. kuruppath,

Thank you for submitting your manuscript to PLOS ONE. After careful consideration, we feel that it has merit but does not fully meet PLOS ONE’s publication criteria as it currently stands. Therefore, we invite you to submit a revised version of the manuscript that addresses the points raised during the review process.

In particular, both reviewers note that there is no positive control for the 10 ms stimulation, an issue that has to be addressed carefully. I also concur that the use of “olfactory identity” might not be appropriate. Stimulating the entire bulb hardly could be seen as conferring olfactory identity. I recommend that the authors stay close to their data and simply state what temporal light stimulus patterns mice can or cannot distinguish. Also, please respond in detail to other concerns raised.

Additional points:

As far as I understand mice are always trained with a 50 ms pulse pattern. Thus it could simply be that mice do not generalize this pattern to the different 10 ms pattern (different “olfactory identity”), but do so to the more similar 25 ms pattern. Hence, what happens if mice are actually trained with a 10 ms pattern?

Please carefully check your spelling as already noted by a reviewer.

Since the authors permanently closed off one leg of the “Y” maze (as they call it) they might as well not call it a “Y” maze since it is not a Y maze. And avoid confusion.

I might have missed it, but it was not clear to me how often during the testing mice were light-stimulated.

We look forward to receiving your revised manuscript.

Kind regards,

Johannes Reisert

Academic Editor

PLOS ONE

Journal Requirements:

2.Thank you for stating the following in the Funding Section of your manuscript:

[This work was supported by the National Institute for Neurodegenerative Disorders and Stroke at

312 the National Institutes of Health intramural program, project number: 1ZIANS003116-01 to LB.]

 [The funders had no role in study design, data collection and analysis, decision to publish, or preparation of the manuscript.]

 b.We note that one or more of the authors is affiliated with the funding organization, indicating the funder may have had some role in the design, data collection, analysis or preparation of your manuscript for publication; in other words, the funder played an indirect role through the participation of the co-authors. If the funding organization did not play a role in the study design, data collection and analysis, decision to publish, or preparation of the manuscript and only provided financial support in the form of authors' salaries and/or research materials, please do the following:

Review your statements relating to the author contributions, and ensure you have specifically and accurately indicated the role(s) that these authors had in your study. These amendments should be made in the online form.

Confirm in your cover letter that you agree with the following statement, and we will change the online submission form on your behalf:

Reviewers' comments:

Reviewer's Responses to Questions

**Comments to the Author**

1. Is the manuscript technically sound, and do the data support the conclusions?

Reviewer #1: Partly

Reviewer #2: Partly

2. Has the statistical analysis been performed appropriately and rigorously? 

Reviewer #1: Yes

Reviewer #2: No

3. Have the authors made all data underlying the findings in their manuscript fully available?

Reviewer #1: Yes

Reviewer #2: Yes

4. Is the manuscript presented in an intelligible fashion and written in standard English?

Reviewer #1: No

Reviewer #2: Yes

5. Review Comments to the Author

Reviewer #1: In this manuscript, Kuruppath and Belluscio investigate the effect of light stimulus duration in mice expressing channelrhodopsin in olfactory sensory neurons during an avoidance behaviour task. Using a Y-maze, mice were trained with both unilateral and bilateral light stimulation paradigms in the olfactory bulb during a foot shock avoidance task. During test sessions their time spent in either the safe or light zone in response to various light stimulation protocols was evaluated.

With this study, the authors aim to address the very timely and exciting question of whether the temporal structure of olfactory information, here focussed on stimulus duration, can be perceived by mammals. The results show a clear difference in behaviour in response to longer (25 ms) and shorter (10 ms) light stimulation when presenting several combinations of unilateral and bilateral light stimulation protocols.

Stimulating the olfactory system artificially with light provides high temporal precision, circumventing the difficulty in precisely controlling odour presentation. However, the lack of information about the level of sensory activation and by sacrificing any specificity regarding glomerular activation patterns impedes the conclusions that can be drawn. In particular, the lack of a positive control showing that the 10 ms light stimulation is sufficient to induce any type of behaviour makes the results hard to interpret.

It is imperative that the authors include a section about the limitations of their experimental design. This should encompass a general discussion about the artificial nature of the stimulus and the constraint of the foot-shock paradigm in the context of olfactory temporal information coding.

Besides my concern about the language that needs significant improvement throughout the manuscript, I listed a number of issues below:

TITLE

In the light of the data presented, the phrase “olfactory identity” is too general for the manuscript title. Please rephrase the title to something that is more descriptive of the actual experiments carried out and the conclusions that can be drawn.

ABSTRACT

The authors need to better highlight the motivation to study the effect of stimulus duration in the context of olfaction and be more specific in the description of experiments. Where is channelrhodopsin expressed? What does the behavioural setup look like? Please also elaborate on the phrase “olfactory information”. The concluding sentence generates high expectations in the reader and should be a lot more specific about the actual experimental results presented in the study and not overstating by generalizing.

INTRODUCTION

Line 27-28: Beyond the contribution of respiration, gaining tight control over the odour stimulus remains a challenge, please elaborate.

Line 28-29: The authors mention complex binding properties in the olfactory epithelium. Please include that natural odours rarely occur isolated and include the potential of antagonistic odour-odour interactions.

Line 31: Please elaborate on the importance of precise timing and include studies that stimulated with actual odours instead of artificial light stimulation (e.g. Parabucki et al., 2019, Jordan et al., 2018).

Line 32: Please add references that provide excellent description of odour plume physics (e.g. by Vergassola, Celani, Crimaldi)

Line 34-35: While all these excellent studies suggest that temporal information is relevant for mammalian behaviour, direct evidence for this is still lacking. This highlights the need for carefully designed behavioural and physiological mammalian experiments. To provide context, the authors should cite some invertebrate studies (some are mentioned in the discussion) to underline the importance of temporal odour information.

Line 39-40: In the context of temporal discrimination in bees the authors should include Szyszka et al., 2013

Line 45: The authors are correct in pointing out the critical effect of sniffing during odour sampling. They should mention that in many studies efforts are made to reduce variability in odour stimulation by triggering the stimulus during a certain phase of the respiration cycle.

Line 48-49: Please mention that both uni- and bilateral stimulation paradigms were performed.

METHODS

Line 93: What is the time between the first training session and the test session, i.e. how many reinforcement sessions did the animals undergo?

Line 95: Why pair foot shock with only unilateral OB stimulation?

Line 98: Why choose ten light 50 ms pulses duration and 150 ms interval? There is no mention of the sniff cycle. While probably out of scope for a revised version of this manuscript, implanting a sniff cannula or use an implanted telemetric sensor would greatly advance the study and open up the opportunity to trigger light stimulation during specific phases of the respiration cycle and, thus allow to relate behaviour to precise stimulus timing and not just duration.

Line 100: Please expand on how the authors chose their stimulus power levels (20-22 mW)? The study lacks any light power titration control experiments to exclude saturated or sub-threshold activation.

RESULTS

Line 134: Please rephrase to “olfactory stimuli” or add references for other modalities.

Line 136: Please elaborate on the direct connection between duration and olfactory identity.

Line 138: Please rephrase to e.g. “…in response to a pattern of 10 light stimuli with a duration of 10 or 25 ms with an inter-stimulus interval (ISI) of 150 ms” to point out that it is not a just single stimulation.

Line 144: Point out that stimulation happens unilaterally.

Line 174: Please include references of mammalian studies to provide context for the challenge of olfactory figure-ground separation.

Figure 2: The data provides the interesting finding that avoidance behaviour can consistently be evoked by the 25 ms but not the 10 ms light pulse protocol. However, there is no control or any explanation whether the 10 ms light pulse at the specific laser power used here is sufficient to activate glomeruli. The fact that the animals avoid both the safe and the light zone upon 25 ms light stimulation is striking and should be discussed in greater detail. Crucially, the authors need to heavily tone down their conclusion that “stimulus duration changes odour identity” since there is no evidence that any glomeruli are activated with the 10 ms light stimulus protocol and, thus no conclusions about odour identity can be drawn.

The data from Figure 3 (unilateral training) and Figure 4 (bilateral training) suggests that bilateral light stimulation is only perceived as such if the stimulus pair is 50-25 ms but not 50-10 ms. Again, there is no proof that the 10 ms light stimulation is sufficient to elicit sensory responses. Hence, the authors’ claim that duration of the bilateral input influences the sensory information needs to be toned down. The phrase “olfactory identity” is very vague and is unwarranted due to the nature of the experiments. Can the authors provide data for, or at least comment on, how animal behaviour would change when switching the light stimulus protocol between the two bulb hemispheres? This would provide confidence that both bulbs can be stimulated similarly and that there is no impairment in light penetration.

In line with the previous results avoidance behaviour was elicited when simultaneously stimulating both bulbs with 25-25 ms but not with 10-10 ms. Again, these experiments do not provide evidence that 10 ms light stimulation elicits any sensory response impeding. One way of resolving this is to provide a positive control that 10 ms light stimulation can elicit avoidance behaviour e.g. by increasing the laser power.

DISCUSSION

Line 281-283: Please rephrase, it is well-established that the number of activated glomeruli increases alongside the odour concentration.

Line 284-285: The authors refer to the influence of flow rate on odour identification. This requires some elaboration on the mechanosensitivity of sensory neurons and projection neurons.

FIGURES

The bar graphs would greatly benefit from a clear label for baseline and light stimulation phases, e.g. by colour code or a legend

The dot colours representing individual animals are distracting

The schematic depicting the stimulation pattern is helpful but it needs clearer labelling, i.e. right and left OB

Figure 2: Point out clearly that stimulation is unilateral

Reviewer #2: Review of PONE-D-20-30938, “The influence of stimulus duration on olfactory identity”

Kuruppath and Belluscio have provided a well-reasoned and thorough study that characterizes how stimulus duration influences odor perception. This is an important topic and in the olfactory system, it is not well-understood how the duration of a stimulus affects how it is perceived. Throughout the study an aversive learning paradigm is combined with optical stimulation of olfactory sensory neurons that express a light-gated cation channel. The most prominent outcome of the study is that even very small changes (< 40 ms) in stimulus duration, fundamentally change information related to odor identity that is carried by olfactory sensory neurons.

In general, the manuscript is clearly written, and the data are well-considered and discussed. The approach using a freely moving behavior coupled with optical stimulation of OSNs is novel; however, there are a few factors that limit enthusiasm, primary among them a lack of a clear control demonstrating that mice are capable of detecting extremely short (10 ms) light pulses. Below I discuss in more detail:

1) My most prominent concern is whether mice are able to perceptually detect 10 ms light pulses. Two recent studies support the feasibility of the approach: Smear et al., 2011 and Li et al., 2014. While the Smear study used nearly twice as much output power, the work of Li and colleagues very nearly matches the methods used here. My concern is that the light-activated cation channel variant used in the present study (ChIEF) is less light sensitive and has a slower opening rate than ChR2 used in other studies. While I do think this point is important for the interpretation of the results in the present study, it can be addressed with added discussion and possible control data. I am curious if there is any data potentially embedded in already collected behavior videos that could help. For example, in the experiments in Figure 2, where mice show no spatial preference to the 10 ms pulses, perhaps the mice freeze, or initiate sniffing behavior when the 10 ms pulse is being delivered. This would lend confidence that OSNs are indeed being activated by the light pulses. It would also need to be shown that mice do not perform a similar behavior in the green light control experiment in Figure 6.

2) Figure 3 & 4: The outcome that bilateral stimulation changes odor identity is reasonable given that different subsets of OSNs will be activated in each hemisphere. However, again, the results here do not lend confidence that mice are able to perceive the 10 ms light pulses. In Figure 3, if mice are trained on unilateral 50 ms light pulses, the test 10/50 ms pulse combination does not change the perception of the ‘odor’. At the same time, the 25/50 ms pulse combination clearly represents a different ‘odor identity’ from unilateral stimulation. A possible explanation is that the 10 ms pulses are not sufficient to activate OSNs in the contralateral bulb, while 25 ms pulses are. The same problem holds true in Figure 4 when mice are trained on bilateral pulses. In this case, if the 10 ms pulses are insufficient to activate the contralateral OSNs, it is not surprising that perception will be different - in effect, only a single hemisphere is contributing to odor identity. Again, this concern can be addressed with some behavioral readout, as mentioned in the point above. Data in Figure 5 with bilateral 10 ms pulses could be helpful.

3) A point that requires some clarification is whether mice are able to explore the third arm of the Y-maize, or whether the spatial location of the animal was a ‘binary’ choice between the light zone and the safe zone. Since the spatial density plots do not show the third arm, I assume this area was unavailable to the animal during testing. The supplemental videos also give this impression. With this in mind, I have concerns about the statistical comparison used. If the mouse is spending less time in the light zone, by design, it must be spending more time in the safe zone. A more appropriate comparison might be between the amount of time spent in the light zone and that same area during the baseline period.

4) In the methods please clarify whether, during the testing phase, mice were stimulated only once with light pulses for each block, or whether they were continuously stimulated each time they enter the light zone.

5) Some discussion is missing regarding how stimulus duration might be influencing perceptual characterization of an odor. While I understand the mechanistic basis of such a phenomenon is beyond the scope of the present study, it might be worthwhile to discuss in the context of psychophysical observations. For example, it is known that long-term odor exposure decreases the perceived intensity of an odor, while the odor quality is not changed. Could it be that your mice are reporting something similar, changes in perceived intensity rather than identity?

Minor:

1) For all figures with heat maps, please use the entire color scale provided. For example, in Figure 2 it appears that none of the spatial density extends into the red portion of the scale. The colors in the plots should be rescaled to utilize the entire color range. Alternatively, a new scale can be provided that more accurately reflects the range of plotted data.

2) Related to the above point, it is not clear whether the heat maps are averages of all 6 mice used, or if the data is from a single animal. If the data is from a single animal, is the same animal used in all four heat maps?

3) Figure 2: The amplitude of the square pluses depicted in parts C and F is different. This makes it appear that less power was used for 10 ms stimulations.

3) Very minor suggestion: The y-axes of the bar plots throughout might be more meaningful if plotted as percent time rather than seconds.

4) Line 143 (and throughout): Please give N values where statistics are present.

5) Line 347: Li et al. is a duplicate citation.

In summary, this submission provides new and potentially important data on how stimulus duration contributes to olfactory sensory perception. The experiments are thoughtfully designed and include an important control to exclude light-induced behavioral artifacts. The most prominent points raised above can potentially be addressed without the need for further experiments, while they may require some reanalysis of existing data. Other points can be addressed by clarification of the methodological details and expanded discussion; therefore, I recommend that the authors are provided the opportunity to submit a revised version of this manuscript.

6. PLOS authors have the option to publish the peer review history of their article (what does this mean?). If published, this will include your full peer review and any attached files.

Reviewer #1: No

Reviewer #2: **Yes: **Joseph D. Zak

---

## [Author Response · Author response to Decision Letter 0]

6 Apr 2021

No specific responses to reviewers

---

## [Decision Letter · Decision Letter 1]

22 Apr 2021

PONE-D-20-30938R1

The influence of stimulus duration on olfactory perception

PLOS ONE

Dear Dr. kuruppath,

Thank you for submitting your manuscript to PLOS ONE. After careful consideration, we feel that it has merit but does not fully meet PLOS ONE’s publication criteria as it currently stands. Therefore, we invite you to submit a revised version of the manuscript that addresses the points raised during the review process.

As also reviewer 2 was confused by the use of the term “Y maze” (which makes 2 out of 3 that evaluated your manuscript) the authors might want to use better terminology. By some mildly wildly extrapolation, it might confuse 66.6 % of the readers of your paper. Please also consider the point raised by Reviewer 2 regarding moving Fig. 5 up to an earlier point in the manuscript. While neither issue will not preclude publication, it will help the reader. I leave it to the authors to decide.

Also note, that Plos One does not copyedit your manuscript. As mentioned in the previous review round, please carefully read your manuscript and also double check your figures. A few examples among many others:

Decide if you are using “olfactory sensory neurons” or “olfactory receptor neurons” and stick with it.“mice line” -> mouse lineIn Fig. 3C and F, do the light exposure monitors need to be exchanged?And see related comments by the reviewers.

We look forward to receiving your revised manuscript.

Kind regards,

Johannes Reisert

Academic Editor

PLOS ONE

Journal Requirements:

Reviewers' comments:

Reviewer's Responses to Questions

**Comments to the Author**

1. If the authors have adequately addressed your comments raised in a previous round of review and you feel that this manuscript is now acceptable for publication, you may indicate that here to bypass the “Comments to the Author” section, enter your conflict of interest statement in the “Confidential to Editor” section, and submit your "Accept" recommendation.

Reviewer #1: All comments have been addressed

Reviewer #2: (No Response)

2. Is the manuscript technically sound, and do the data support the conclusions?

Reviewer #1: Yes

Reviewer #2: Yes

3. Has the statistical analysis been performed appropriately and rigorously? 

Reviewer #1: Yes

Reviewer #2: Yes

4. Have the authors made all data underlying the findings in their manuscript fully available?

Reviewer #1: Yes

Reviewer #2: Yes

5. Is the manuscript presented in an intelligible fashion and written in standard English?

Reviewer #1: Yes

Reviewer #2: Yes

6. Review Comments to the Author

Reviewer #1: Fig. 5I: The schematic for bilateral stimulation is the same as for 5F (although squeezed horizontally) and thus of limited helpfulness without digging into the figure legend. The authors should adapt to assist the reader through the figure.

Quite a lot of grammatical inaccuracies and typos can be found throughout the manuscript. The paper would greatly benefit from an additional round of proof reading.

Reviewer #2: Comments for revised manuscript PONE-D-20-30938, The influence of stimulus duration on olfactory perception.

This work has been improved by the addition of new control data and attention to the comments of both reviewers. However, there are a few minor points that I still contend in this revised version. I hope that the authors will agree to consider them.

1) I agree with the editor's comments that calling this task a Y-maize is inappropriate given that that the task used is more analogous to a two-chamber forced choice. It is mentioned in the response to the reviewers that readers might find it confusing to change the task description; however, this reader was indeed confused by the current description as a Y-maize. I strongly encourage the authors to modify this in the text. Perhaps, in the methods, the authors could describe that a Y-maize was used to make a two-chamber task, and use “two-chamber” throughout the text.

2) I am unconvinced by the response regarding figures. It seems to be little trouble to, as Reviewer 1 suggested, add “a clear label for baseline and light stimulation phases.” Similarly, I disagree that the full range of the color scale is being used in all figures. It appears that <25% of the range is being covered for most data. I understand that these may be software-generated figures, but the data should be optimally displayed. My suggestion is to either change the scale bar or resale the data to match the axis. The point is made in the response to the reviewers – which points out that Figure 5B is properly scaled. Why not the rest of the data?

3) It might be helpful to move the new control data (Figure 5G-I) to earlier in the manuscript. The question of whether the mice can detect 10 ms light pulses arises following the very first data (Figure 2) and goes unmentioned following Figures 3-4, were similar questions come up. I understand including the 10 ms data with a similar experimental approach, but this unanswered question distracts from the rest of the discussion.

4) Lastly a nomenclature issue. ChIEF is not synonymous with ChR2. Throughout the text ChR2 should be replaced when what is really meant is ChIEF.

Overall, I feel that this work has merit, value, and is much improved after revision. I also hope that authors will address these rather limited points.

7. PLOS authors have the option to publish the peer review history of their article (what does this mean?). If published, this will include your full peer review and any attached files.

Reviewer #1: No

Reviewer #2: No

---

## [Author Response · Author response to Decision Letter 1]

22 May 2021

No specific responses for reviewers and editor

---

## [Editor Report · Decision Letter 2]

26 May 2021

The influence of stimulus duration on olfactory perception

PONE-D-20-30938R2

Dear Dr. kuruppath,

We’re pleased to inform you that your manuscript has been judged scientifically suitable for publication and will be formally accepted for publication once it meets all outstanding technical requirements.

Kind regards,

Johannes Reisert

Academic Editor

PLOS ONE
---

## [Editor Report · Acceptance letter]

1 Jun 2021

PONE-D-20-30938R2 

The influence of stimulus duration on olfactory perception 

Dear Dr. Kuruppath:

I'm pleased to inform you that your manuscript has been deemed suitable for publication in PLOS ONE. Congratulations! Your manuscript is now with our production department. 

Kind regards, 

on behalf of

Dr. Johannes Reisert 

Academic Editor

PLOS ONE